

# *DChIPRep*, an R/Bioconductor package for differential enrichment analysis in chromatin studies

Christophe D. Chabbert[1,2], Lars M. Steinmetz[1,3,4] and Bernd Klaus[1]

[1] Genome Biology Unit, European Molecular Biology Laboratory, Heidelberg, Germany
[2] Oncology iMed, CRUK-Cambridge Institute, Astra Zeneca, Cambridge, United Kingdom
[3] Stanford Genome Technology Center, Stanford University, Palo Alto, California, United States
[4] Department of Genetics, Stanford University School of Medicine, Stanford, California, United States

## ABSTRACT

The genome-wide study of epigenetic states requires the integrative analysis of histone modification ChIP-seq data. Here, we introduce an easy-to-use analytic framework to compare profiles of enrichment in histone modifications around classes of genomic elements, e.g. transcription start sites (TSS). Our framework is available via the user-friendly R/Bioconductor package *DChIPRep*. *DChIPRep* uses biological replicate information as well as chromatin Input data to allow for a rigorous assessment of differential enrichment. *DChIPRep* is available for download through the Bioconductor project at http://bioconductor.org/packages/DChIPRep.
**Contact.** DChIPRep@gmail.com.

## INTRODUCTION

The elementary component of eukaryotic chromatin, the nucleosome, is composed of 147bp DNA fragments wrapped around an octamer comprising two copies of four of the histone proteins. The N-terminal tails of these proteins are subject to multiple post-translational modifications (PTM) including acetylation, phosphorylation and methylation (*Lawrence, Daujat & Schneider, 2016*). Such modifications may be found in combination on several residues of the same histone proteins, adding to the complexity of the combinatorial space of PTM that may be explored (*Tan et al., 2011*). Recent studies have highlighted the importance of these PTM in key cellular processes such as transcription, DNA replication and repair. Protocols based on chromatin immunoprecipitations followed by deep sequencing (ChIP-seq) allow for a genome-wide mapping of these modifications. Such endeavors have resulted in the generation of complex sequencing datasets that require appropriate bioinformatics tools to be analyzed. From this data, profiles of enrichment in histone modifications around classes of genomic elements, e.g. transcription start sites (TSS) are routinely computed. Once these enrichment profiles have been obtained, a common analysis task is to compare them between experimental conditions. However, due to a lack of tools tailored to the

Corresponding author
Bernd Klaus, bernd.klaus@embl.de

assessment of differential enrichment, these comparisons are often performed in a purely descriptive manner (e.g. by comparing plots of enrichment profiles around transcription start sites). In this article, we present a workflow to assess differential enrichment in a statistically rigorous way. This workflow is implemented in a user-friendly package named *DChIPRep* that is available via the Bioconductor project (*Huber et al., 2015*).

## Review of existing tools and approaches

Several software tools designed to analyze certain aspects of histone modification data are already available. These usually focus on one or several of the 3 main aspects explored in chromatin biology: the genome-wide determination of nucleosome positions (not adressed by DChIPRep), the identification of genomic loci enriched in the modifications of interest (so-called peaks, not addressed by DChIPRep) and differential binding analysis, an aspect tackled by our package. Diverse statistical and numerical approaches have been concurrently implemented to infer nucleosome positions, including Fourier transform (*nucleR*, *Flores & Orozco, 2011*), Gaussian filtering (*Genetrack*, *Albert et al., 2008*), wavelets (*NUCwave*, *Quintales, Vázquez & Antequera, 2014*) as well as probabilistic or Bayesian approaches (*NucleoFinder Becker et al., 2013*, *PING 2.0 Woo et al., 2013*, *NOrMAL Polishko et al., 2012*). Mutiple approaches based on signal smoothing and local background modeling have also been implemented to identify regions with high numbers of mapped reads (peaks) of variable width (*Feng et al., 2012*; *Bailey et al., 2013*).

Some algorithms proposed recently go beyond the determination of nucleosome or peak positions and aim at assessing differential enrichment. However, they commonly rely on the identification of regions of interest (e.g. around called peaks) using the ChIP-seq datasets themselves e.g. *DiffBind*, (*Ross-Innes et al., 2012*; *Stark & Brown, 2011*). Notably, *csaw* (*Lun & Smyth, 2014*) allows for a genome wide identification of differential binding events without an a priori specification of regions of interest. It uses a windowing approach and implements strategies for a post hoc aggregation of significant windows into regions. Although *DESeq2* is commonly used for differential binding analysis of ChIP-Seq data (*Bailey et al., 2013*), to the best of our knowledge, no direct approach to compare enrichment profiles of histone modifications around classes of genomic elements exists so far. Furthermore, most existing tools do not offer the possibility to directly correct for biases using the Input chromatin samples. Commonly, these profiles are analyzed in a purely descriptive manner and conclusions are drawn solely from plots of metagenes/metafeatures (e.g. transcription start site plots).

Here we present *DChIPRep*, an R/Bioconductor package designed to compute and compare histone modification enrichment profiles from ChIP-seq datasets at nucleotide resolution. The workflow implemented in *DChIPRep* uses both the biological replicate and the chromatin Input information to assess differential enrichment. By adapting an approach for the differential analysis of sequencing count data (*Love, Huber & Anders, 2014*), *DChIPRep* tests for differential enrichment at each nucleotide position of a metagene/metafeature profile and determines positions with significant differences in enrichment between experimental groups. An overview of the complete workflow is given next.

## Overview of the implemented framework

The framework implemented in *DChIPRep* consists of three main steps:

1. The chromatin Input data is used for positionwise-normalization.
2. The methodology of *Love, Huber & Anders (2014)* is used to perform positionwise testing. A minimum absolute $\log_2$-fold-change greater than zero between the experimental groups is set during the testing procedure to ensure that called positions show an non-spurious differential enrichment.
3. Finally, in order to assess statistical significance, local False Discovery Rates (local FDRs, *Strimmer, 2008*) are computed from the p-values obtained as a result of the testing step. Local FDRs assess the significance of each positions individually and are thus well suited for the detection of fine-grained differences.

## Real data analysis

We first apply *DChIPRep* and a modified version of its framework using methodology inspired by the *csaw* and *edgeR* (*Lun & Smyth, 2014*; *McCarthy, Chen & Smyth, 2012*) packages to yeast ChIP-seq data and compare the enrichment profiles around TSS in wild-type and mutant strains, demonstrating how our package can derive biological insights from large-scale sequencing datasets.

We furthermore analyze a published mouse data set by *Galonska et al. (2015)*, to compare H3K4me3 enrichment around selected TSS in embryonic stem cells grown in two conditions (serum/LIF and 2i conditions).

# METHODS

## General architecture of the package

*DChIPRep* uses a single class `DChIPRepResults` that wraps the input count data and stores all of the intermediate computations. The testing and plotting functions are then implemented as methods of the `DChIPRepResults` object. The plotting functions return *ggplot2* (*Wickham, 2009*) objects than can subsequently be modified by the end-user.

DChIPRep's analytical method uses histone modification ChIP-Seq profiles at single nucleotide resolution around a specific class of genomic elements (e.g. annotated TSS). In the case of paired-end reads originating from chromatin fragmented using microccocal nuclease (MNAse), such profiles can be obtained using the middle position of the genomic interval delimited by the DNA fragments (Fig. 1).

Thus, the variables characterizing the samples are the genomic positions relative to a specific class of genomic elements (e.g. TSS). These variables take the values given by the number of sequenced fragments with their center at these specific positions. The data is summarized across genomic features (e.g. genes or transcripts) at each of these nucleotide positions, so that metagene/metafeature profiles are obtained. The input data for DChIPRep can be alignment files in the SAM format or already processed count data.

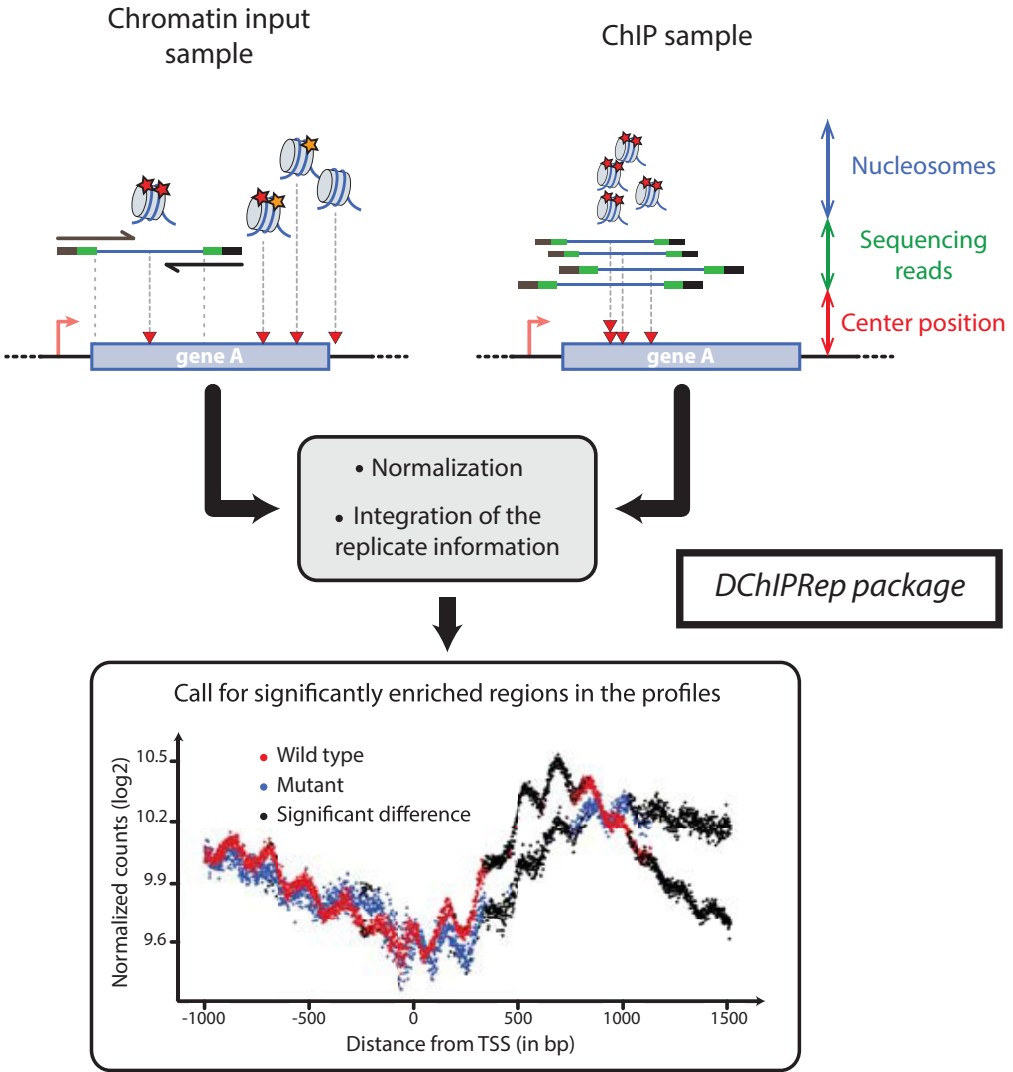

**Figure 1 Illustration of the *DChIPRep* workflow.** Chromatin Input- and ChIP-data are analyzed jointly and positions showing significantly different enrichment are identified using the replicate information.

## Data import

DChIPRep has two possible data input formats. The input data can be two count tables per sample (for ChIP and Input), with the genomic features used (e.g. genes or transcripts) in the rows and the position wise counts per genomic feature in the columns. Alternatively, one can provide two count tables for ChIP and Input that contain the data at the metafeature level, such that the data is summarized across individual genomic features. These tables then have one row per position relative to the genomic element (e.g. TSS) studied and one column per sample.

DChIPRep can either import tab-separated .txt files (two files per experimental sample with data at the level of the individual genomic features) or two R count matrices for ChIP and Input data, which contain the data already summarized at a metafeature level (summarized across features per position). A table containing the experimental

conditions and other sample specific annotation is needed as well. A Python script (DChIPRep.py) is also provided along with the package to generate suitable tab-separated input files from SAM alignment files of paired-end data and a gff annotation. The script relies on the HTSeq module (*Anders, Pyl & Huber, 2015*) and may be customized via multiple parameters.

It also possible to directly import .bam files using the function `importData_soGGi`. The function also supports the import of single-end data and uses functionality from the soGGi package (*Dharmalingam & Carroll, 2015*). The resulting imported data is then a matrix of metafeature profiles.

Further details on the data import can be found in the package vignette, which is available via Bioconductor and as Supplemental Information 1.

## Computation of the metafeature profile

Once the data is summarized on the feature level (i.e. count tables), we can compute the metafeature profiles with the function `summarizeCountsPerPosition` for each of the ChIP-Seq and chromatin Input samples.

The function first filters out features with very low counts. Then, in order to summarize the data across features, a trimmed mean of the counts at each position is computed.

Finally, these positionwise mean values are multiplied by the number of features retained at each position. This way, a raw metafeature profile for each individual sample is obtained.

## Call for enriched regions

The statistical approach implemented in *DESeq2* is used to call for significantly deferentially enriched positions (*Love, Huber & Anders, 2014*).

Here, the chromatin input is used to compute normalization factors that correct for potential local biases in chromatin solubility, enzyme accessibility or PCR amplification.

Essentially, we use the positionwise ratios of ChIP and Input data as the normalized data: we adjust the total counts (representing the library size) of the chromatin Input metafeature profiles so that they match the library size of their corresponding ChIP samples. This corrects for global differences in sequencing depth between the ChIP and Input samples and is commonly performed as a simple scaling normalization in the analysis of high-throughput sequencing data as exemplified in *edgeR*. The adjusted Input data is now on the same scale as the ChIP data and can directly be used as a position-specific normalization factor in *DESeq2*. Other reports have also shown that it is possible to use the Input data directly as an "ordinary" sample and then test for differences between ratios of ratios (*Love, 2014*)—this functionality might be added as an option in a future version of the package.

After specifying a minimum fold change, Wald tests are performed to assess significant changes in the metagene/metafeature profiles.

Finally, local FDRs estimated by the `fdrtool` (*Strimmer, 2008*) package are used to assess statistical significance based on the p-values obtained from the Wald-test.

All of these steps are implemented in the `runTesting` function (Fig. 1).

## Plotting functions

DChIPRep provides two plotting functions to represent and inspect the final results of the analysis. The `plotProfiles` function summarizes the biological replicates by taking a robust position wise mean (*Hampel, Hennig & Ronchetti, 2011*) and then plots a smoothed enrichment profile around the genomic element class of interest (e.g. TSS).

The `plotSignificance` function plots the unsmoothed enrichment profile and highlights positions with a significant difference in enrichment as returned by the runTesting function (Fig. 1). The plotting functions return *ggplot2* objects that can be easily customized.

While these plotting capabilities are relatively elementary, we would like to point out that the Bioconductor package *soGGi* as well as the standalone tool *ngs.plot* (*Shen et al., 2014*) offer more sophisticated visualizations of ChIP-seq profiles, which are not at the core of DChIPRep functionality.

# RESULTS

## A yeast case study

We applied *DChIPRep* to a paired-end ChIP-seq dataset for which biological replicates are available (*Chabbert et al., 2015*). This dataset also includes sequences from the associated chromatin inputs, which were obtained using MNAse digestion. For this particular case study, we have only selected the sequencing data generated using a classical ChIP-seq protocol (while other samples have been profiled using a different protocol in the same publication). Using the annotation from *Xu et al. (2009)*, we compared the enrichment of the H3K4me2 mark in annotated ORFs (5,170 items) in the wild type strain of Saccharomyces cerevisiae and the *set2*Δ mutant. We have called a significant enrichment (local FDR < 0.2) in the mutant for 906 positions located within 1,500 bp downstream of the transcription start site (Fig. 1).

## Steps for a typical analysis

In order to illustrate the usage of *DChIPRep* we document the series of simple commands that are needed to be to run a typical analysis.

After the data has been proccesed, we first need to import a table that contains the annotation information for our samples. This table contains information on the count table file names and the desired number of up- and downstream positions to be compared, as well as the experimental group a sample belongs to. As mentioned above, details on the required format of the annotation table can be found in the package vignette in Supplemental Information 1.

We can then import the data using the function `importData`.

---

**Listing 1  Data Import**

```
sampleTable_K4me2 <- read.csv ("sampleTable_K4me2.csv")
importedData <- importData (sampleTable_K4me2)
```

---

After then data import, we can perform the positionwise testing with the `runTesting` function, extract the results using the `resultsDChIPRep` function and finally obtain the significance plot in Fig. 1 via a call to the `plotSignificance` function.

---

**Listing 2 Results and Figure**

```
testResults <- runTesting(importedData)
testResults <- resultsDChIPRep(testResults)
plotSignificance(testResults)
```

---

## A comparison to an *csaw*/*edgeR*-based pipeline

The framework implemented in *DChIPRep* uses the *DESeq2*-package (*Love, Huber & Anders, 2014*) to perform the statistical testing. The *csaw*-package (*Lun & Smyth, 2014*) implements a strategy based on methods implemented in *edgeR* (*McCarthy, Chen & Smyth, 2012*) to assess differential binding in ChIP-Seq data sets genome-wide. While *csaw* and *DChIPRep* are not directly comparable, we can adapt the *csaw* framework to assess the differential enrichment (for a summary of the *csaw* framework, see section 1.3 of the *csaw* user guide at Bioconductor).

Specifically, we used the log-normalization factors computed from the chromatin-input as offsets for the GLM-model and then applied the quasi-likelihood (QL) methods of *Lund et al. (2012)* to perform a dispersion shrinkage and an appropriate F-test to assess the differential enrichment. Note that since *edgeR* does not allow for an a priori specification of a fold change threshold, we had to specify it post hoc. The complete analysis can be found in Supplemental Information 2.

Figure 2 shows the results of this approach. The modified pipeline identified 1,171 positions as significantly deferentially enriched located within 1,500 bp downstream of the transcription start site. Comparing Figs. 2 to 1, we see that the *edgeR*-based pipeline gives very similar results in this case. In fact, all 906 positions called by *DESeq2* downstream of the TSS are also called by *edgeR*. This indicates that the local FDR based thresholding, which assesses the significance of the position individually is more important than the actual statistical test performed. We still observe that *DChIPRep* identifies deferentially enriched regions more consistently, while the *edgeR* pipeline calls many positions with moderate fold-changes up to 250 p downstream of the TSS as significant. This might be due to the fact that a post hoc fold change thresholding had to be performed. *DChIPRep* would therefore be less prone to calling false positive as it is less sensible to weak enrichment (which might be resulting from intrinsic variability in the performance of immunoprecipitation for example).

## Analysis of H3K4me3 profiles in mouse embryonic stem cells (ESCs)

As an additional case study we analyzed the H3K4me3 enrichment profiles of mouse ESCs from *Galonska et al. (2015)*. The data consists of two replicates in two conditions that correspond to a typical stem cell culture conditions (serum/LIF) and a 24h-2i

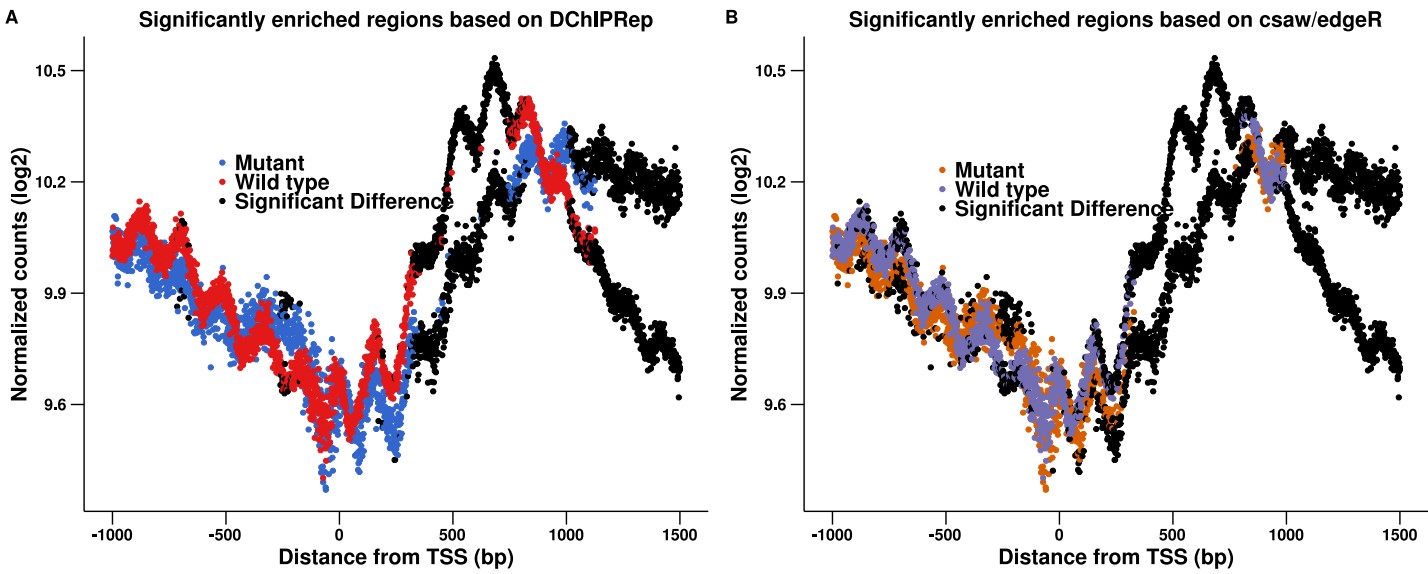

**Figure 2** Results of the *csaw/edgeR*-based calling for enriched regions are shown in (**B**). The results presented in Fig. 1 are shown again in (A). We applied an *edgeR*-based testing to the data (instead of using *DESeq2*). This included a post hoc thresholding of the fold-changes. The figure shows that this pipeline calls many positions with moderate fold-changes up to 250 p downstream of the TSS as significant.

condition characterized by two inhibitors (2i) of the MEK and GSK3 pathways respectively (thought to represent an embryonically restricted ground state). As only one chromatin Input was sequenced (whole cell extract, WCE), we use this data as Input data for all four samples.

We downloaded the data from the short read archive (SRA) at the European Nucleotide Archive (ENA, accession PRJNA242892) and the lists of called peaks in two conditions from GEO (GSE56312).

The raw single-end reads were aligned to the mm9 reference genome using bowtie2 (*Langmead & Salzberg, 2012*) with default options. Then, filtering of unmapped, low mapping quality (< 10), duplicated and multi-mapping reads was performed using Picard tools (*Broad-Institute, 2016*). We then inferred the fragment length for each sample using using cross correlation plots from SPP (*Kharchenko, Tolstorukov & Park, 2008*). We subsequently merged the two peak lists provided by the authors (GEO GSE56312) and then annotated the peaks to the closest mm9 TSS using the function *annotatePeakInBatch* from the package *ChIPpeakAnno* (*Zhu et al., 2010*; *Zhu, 2013*). We finally used the function *regionPlot* from the *soGGi* package (*Dharmalingam & Carroll, 2015*) to create metagene profiles around this subset of TSS that are close to the peaks identified by the authors of the original study. We kindly refer the reader to the preprocessing-code in Supplemental Information 3 for further information.

Figure 3 shows that the profiles in both conditions are quite smooth and very similar. We therefore used *DChIPRep* with a log fold change threshold of 0 and a local FDR threshold of 0.3 to identify significantly different positions. The serum condition has a somewhat higher enrichment near the TSS, while the 24h-2i condition is slightly more

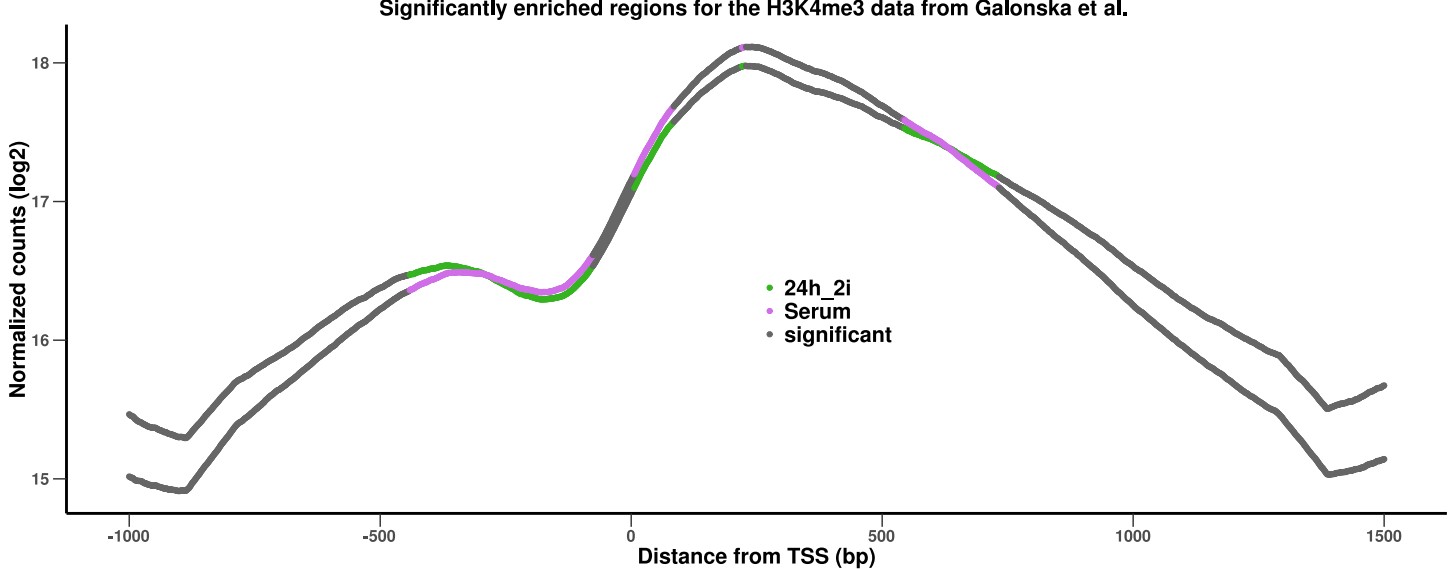

**Figure 3** Results of the analysis of the H3Kme3 data from *Galonska et al. (2015)*. The profiles are based on TSS close to identified peaks of the histone modification in the two conditions.

enriched upstream and downstream of the TSS. The genomic regions where both profiles become very similar in shape and intensity are correctly identified as not significantly differentially enriched. While the overall differences are subtle, *DChIPRep* successfully and correctly identifies regions with differences, provided suitable parameters are used.

### Reproducible research

The complete code and the data used for the case study can be found in Supplemental Information 2.

## DISCUSSION AND CONCLUSION

The package *DChIPRep* provides an integrated analytical framework for the computation and comparison of enrichment profiles from replicated ChIP-seq datasets at nucleotide resolution or lower.

Starting from the primary alignment of paired-end reads, the software allows a rapid identification of significantly differentially enriched positions relative to classes of genomic elements and provides straightforward plotting of the enrichment profiles.

We also applied the *DChIPRep*-package to two published data sets using yeast and mouse a model systems. The yeast case study demonstrates *DChIPRep*'s favourable performance when compared to a pipline inspired by the *csaw*-package for differential binding analysis.

## ACKNOWLEDGEMENTS

We thank Sophie Adjalley, Vicent Pelechano, Aleksandra Pekowska and Alejandro Reyes for helpful discussions and critical comments on the manuscript.

### Funding

This work was supported by a PhD fellowship from Boerhinger Ingelheim Fonds (to C.D.C); and the Deutsche Forschungsgemeinschaft (1422/3-1 to L.M.S.). The funders had no role in study design, data collection and analysis, decision to publish, or preparation of the manuscript.

### Competing Interests

The authors declare that they have no competing interests.

### Author Contributions

- Christophe D. Chabbert conceived and designed the experiments, performed the experiments, analyzed the data, wrote the paper, prepared figures and/or tables, reviewed drafts of the paper.
- Lars M. Steinmetz conceived and designed the experiments, reviewed drafts of the paper.
- Bernd Klaus conceived and designed the experiments, performed the experiments, analyzed the data, wrote the paper, prepared figures and/or tables, reviewed drafts of the paper.

### Data Deposition

Raw data and analysis code to reproduce the paper figures is available as Supplemental Material.

### Supplemental Information

Supplemental information for this article can be found online at http://dx.doi.org/10.7717/peerj.1981#supplemental-information.

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
