# Peer review of "DChIPRep, an R/Bioconductor package for differential enrichment analysis in chromatin studies"

_PeerJ, doi:10.7717/peerj.1981_

## Round 0.1 · original submission · Major Revisions

The reviewers have raised a number of technical issues that will need to be addressed. It is also important to reference the related tools that the reviewers have mentioned and discuss any conceptual similarities or differences with these. At the same time, please note that novelty (and/or "superiority" of your software versus existing tools) is not a requirement for acceptance of your work to PeerJ. Likewise, analyses suggested by reviewers that would entail a significant extension of the paper's scope are at your discretion.

·

Basic reporting

The authors present a short paper describing a new R package called DChIPRep for differential analysis of ChipSeq data including replicates. The package is in practice a wrapper on the typical pipeline where the researcher tests for differential enrichment in pre-defined regions using HTSeq and Deseq2. This is a common practice, however to my knowledge there was no dedicated package in R to do that, so it is likely to be useful for experimentalists who do not want to program their own pipeline using HTseq and Deseq2.
While the paper itself is short, and serves mostly as an extended vignete for the package, I think it has serious problems that need to be addressed before publication. 

Major points:
- The article cnstantly mixes software for nucleosome position detection (such as nucleR) and for ChIPSeq analysis (such as DiffBind). For example The authors write "Several software tools designed to analyze certain aspects of histone modification data are already available. They mostly focus on genome wide determination of nucleosome positions..." I understand that this might be connected to the background of the research for which the package was developed, however it is not an accurate desctription of the field. The whole section should be rewritten to properly discern between the question of nucleosome positioning (_not_ addresed by this method), chip-peak calling (_not_addresed by this method) and differential chip-seq analysis (addresed by this method).
- The authors use for their case study a dataset from their recent paper on the BAR-Chip method, which uses chip-seq on the mnase digested and barcoded material. They sometimes refer to this dataset as MNase-Seq (e.g. line 95 or line 146) and sometimes as CHIP-data (e.g. in Fig. 1 legend). I understand that both are adequate, but it should be clarified (I had to look it up in the referenced paper, because typically these two procedures are not performed together). As I mention in the "experimental design" section, I think it would be much better to use a standard chip-seq method altogether, but if the authors want to use their own complex data, they should present it properly.
- The practice of using DeSeq or DeSeq2 statistics for differential ChipSeq is rather common. For example DeSeq is mentioned for this purpose in the paper on "Practical Guidelines for the Comprehensive Analysis of ChIP-seq Data" by T. Bailey and other ( PLoS Comp. Bio. 2013). The authors should acknowledge that this approach is rather common, which makes the automated tool even more welcome, rather than stating that "however, to the best of our knowledge, no direct approach to compare (...) exists so far."
Minor points:
- In the introductory sectionwhen the authors discuss PTM of histone tails, it would be useful to give a reference to some material where a reader can actually get some more information if they indeed don't know ahat these are. I would also mention that the modifications can be done to multiple residues making it a much more complex system.
- line 76 - the second step of the framework states that the "fold-change greater than zero" is set, which I assume describes the Wald test, however the description here is lacking. Does the statement imply that a one-sided test is performed? or is the fold-change here with respect to the input? It should be clarified.
- in the Data Import section, the authors mention that they provide a DChIPRep.py utility that extracts count data. However they do not mention it relies on the HTseq package . This is an important dependency as the .py file fails if there is no HTseq installed. This should be mentioned in the text and the HTseq tool should be referenced.
- line 59. there is a word "peer" that seems out of context

Experimental design

Major points:
- The case study focuses on the BAR-CHIP data from Chabbert et al 2015. This is a recently developed and new protocol. Not exactly a standard in the field. If the method should be applicable to the wide spectrum of applications I would suggest using it on any, publicly available standard Chip-Seq dataset. The authors can present it also on their data to potentially highlight some additional features, however the title and abstract suggest that the tool is generally applicable to "chromatin studies" and "chip-seq data" and this is not demonstrated.
- The additional use of Wald test is not well supported. This test assumes Normality of data, which is rarely satisfied in NGS data. It's not clear to me why this is necessary and whether there is evidence for normality of the metagene profile differences. I'd suggest a reference to a study with relevant statistical data
- The supplementary dat allows for replication of the plots, however, the instruction is misleading . The data is provided as an RData file, whiel the instruction is based on the assumption that the counts will be read from text files. This should be changed.

Validity of the findings

no comments

Additional comments

I think this package can be very useful to many people, however the paper needs substantial modifications to both description of the method as well as the case study.

·

Basic reporting

This article presents a Bioconductor package wrapper tool for differential ChIP analysis fitted to metagene profiles.
The article does make mention of the existence of tools for the generation or analysis of metagene profiles but fails to make appropriate referencing to these tools despite referencing the DiffBind and CSAW Bioconductor package.
I would expect to see references to NGSplot (as a leading package in metaprofile related count table generation) and to the soGGi Bioconductor package released previously (of which I am author and which has a very high overlap of methods/approaches detailed below.).
Appropriate referencing of NGS plot and soGGI is required to provide sufficient background to previously available tools sharing highly overlapping functions.

Experimental design

The package has significant overlap with pre-existing tools but offers a wrapper to the visualisation and reporting of positions differential for ChIP signal.

This package however provides no method of generation of metaprofiles in R despite many R packages providing methods to generate count tables from single-end ChIP-seq. The external python script provided will only work on SAM files (citing some comment on standards in BAM format on their github page I dont understand). The use of SAM files would greatly inflate the size of the input files.

It is unclear whether their python tool works for single end ChIP-seq to me. From a quick review of the python code it appears to rely on paired-end data. Since the vast majority of ChIP-seq is single end, the use of this tool is very limited. Should this be the case then this limitation must be mentioned in the text.

The high overlap with much of the functionality already seen within soGGi should be addressed with a focus on how DChIPRep differs.

soGGi already carried the functionality seen within DChIPRep.py and implemented this within Bioconductor framework. This includes selection of pairs of read satisfying minimum and maximum fragment lengths and the filtering for duplicate reads. soGGi previously offers much of the functionality of the main package including the use of trimmed means, reporting plots as GGplot for user-customisation and the construction of summarisedExperiment objects for ease of use with differential tools in a few lines.

In order to offer a useful tool, a complete workflow would be required working from single-end (including fragment length estimation/extension) or paired-end BAM files to visualisation of differential regions. I believe this could have been resolved for example by taking a dependency on soGGi and using code as below (here for paired data allowing for fragment length selection).

# Where mm9TSS is TSS locations in a GRanges object.
mnaseData <- regionPlot("myMNAse.bam",mm9TSS,style="point",distanceUp = 2000,distanceDown = 2000,distanceAround = 2000,format="bam",paired=T,removeDup=F,minFragmentLength=100,maxFragmentLength=150)
myBPcountTable <- assay(mnaseData)
rownames(myBPcountTable) <- rowRanges(mnasData)$name
colnames(myBPcountTable) <- colnames(mnasData)

Although it is clearer in the packages code, I would like to see a more detailed description of the generation of normalisation factors for input correction including any references to better assess the methodology.

My major concern is that in its current state, this package does not offer original functionality and as a pipeline package fails to offer a complete workflow.

Validity of the findings

This package's performance in detecting differentially bound regions is visually compared to that implemented in CSAW package. This comparison appears to show very similar results with the authors suggesting differences are largely due to the application of the fold change filter. This suggests that this wrapper tool offers little advantage over a similar method previously implemented.
A direct comparison of the methods in a table/vennDiagram of common versus unique differential regions and/or presentation of CSAW and DiffChIPRep results in a single plot or side-by-side would help clarifying how different these are and where differences lie.

---

## Round 0.2 · accepted · Accept

Thank you for addressing the reviewers comments. I am happy to accept the revised manuscript.

·

Basic reporting

No comments

Experimental design

No comments

Validity of the findings

no comments

Additional comments

I'm fully satisfied with the changes made by the authors.